# Synthesis of Cellulose–Poly(Acrylic Acid) Using Sugarcane Bagasse Extracted Cellulose Fibres for the Removal of Heavy Metal Ions

**DOI:** 10.3390/ijms24108922

**Published:** 2023-05-18

**Authors:** Fuchao Li, Zhemin Xie, Jianfeng Wen, Tao Tang, Li Jiang, Guanghui Hu, Ming Li

**Affiliations:** College of Science & Key Laboratory of Low-Dimensional Structural Physics and Application, Education Department of Guangxi Zhuang Autonomous Region, Guilin University of Technology, Guilin 541004, China

**Keywords:** cellulose, modified cellulose hydrogels, heavy metal ions, graft copolymer, sugarcane bagasse

## Abstract

In this study, sugarcane bagasse (SCB) was treated with sodium hydroxide and bleached to separate the non-cellulose components to obtain cellulose (CE) fibres. Cross-linked cellulose–poly(sodium acrylic acid) hydrogel (CE–PAANa) was successfully synthesised via simple free-radical graft-polymerisation to remove heavy metal ions. The structure and morphology of the hydrogel display an open interconnected porous structure on the surface of the hydrogel. Various factors influencing batch adsorption capacity, including pH, contact time, and solution concentration, were investigated. The results showed that the adsorption kinetics were in good agreement with the pseudo-second-order kinetic model and that the adsorption isotherms followed the Langmuir model. The maximum adsorption capacities calculated by the Langmuir model are 106.3, 333.3, and 163.9 mg/g for Cu(II), Pb(II), and Cd(II), respectively. Furthermore, X-ray photoelectron spectroscopy (XPS) and energy-dispersive X-ray spectrometry (EDS) results demonstrated that cationic exchange and electrostatic interaction were the main heavy metal ions adsorption mechanisms. These results demonstrate that CE–PAANa graft copolymer sorbents from cellulose-rich SCB can potentially be used for the removal of heavy metal ions.

## 1. Introduction

Water pollution has become a global problem and is one of the most devastating economic issues in recent decades, especially pollution by heavy metal ions [1,2]. The primary sources of heavy metal ions in water include electroplating, batteries, dyes, metallurgy and smelting, chemical manufacturing, and the electrical industry [3,4]. Water pollution from heavy metal ions poses a serious risk to human health and the ecological environment because of their high toxicity and non-biodegradability [5]. Therefore, the removal of heavy metal ions from polluted water is crucial. Many techniques are currently available to treat heavy metal ions, including ion exchange, chemical precipitation, electrolysis, membrane separation, and adsorption [6,7]. Of these, adsorption is among the most economical, simple, and effective methods for water purification [8].

Sugarcane bagasse (SCB) cellulose-based nanomaterials are green materials used to prepare composite materials and heavy–metal adsorbents [9,10,11]. Because of the high cellulose (CE) content in bagasse, alkali pre-treatment followed by bleaching is used to remove other components, such as lignin [12], thus producing SCB fibres. Plant-derived cellulosic raw materials have limited adsorption capacity; therefore, cellulose must be chemically modified to achieve an efficient adsorption capacity [13,14]. The obtained cellulose sulphonation could remove Fe^3+^, Pb^2+^, and Cu^2+^ in an orderly manner with good selectivity and high efficiency [15]. In addition, the use of acid anhydride-modified cellulose significantly increased its adsorption capacity (172.5 mg/g) [16]. Cellulose-based adsorbents have many advantages, such as low cost-effectiveness, non-toxicity, and high adsorption capacity [17,18].

Hydrogels can be used to remove organic and inorganic pollutants from water/wastewater owing to their unique properties, such as low density, 3D interconnected porosity [5,19], high surface area, and exceptional wettability [20,21,22]. In particular, the existence of a large number of active adsorption sites and interconnected porous structures gives gels great potential for application in heavy metal ion adsorption [23,24]. Polyacrylic acid (PAA) is an environmentally friendly polymer, and a PAA hydrogel has been used as an adsorbent [25,26]. Chen et al. studied the modification of bentonite with sodium polyacrylate (SPB), which exhibited high adsorption of heavy metals [27]. The hydroxyl groups on cellulose can be directly chemically functionalised [28]. Through grafting and cross-linking, they can be combined with the hydrogel network to form an overall stable cross-linked network, which can enhance the stability of the modified cellulose materials under different conditions [6,13,17,29]. For example, cellulose extracted from pineapple leaves was used to synthesise cellulose-g-PAA with high water absorption [30]. Guleria et al. used hydrogen peroxide and ascorbic acid as initiators and graft–copolymerised cellulose-rich biomass with acrylamide and acrylic acid as a monomer mixture to form a hydrogel that dramatically removes heavy metal ions [17].

In this study, SCB was chemically treated with alkaline hydrogen peroxide to remove the non-cellulosic components. The extracted CE was graft-copolymerised with acrylic acid (AA) using potassium persulfate (KPS) and N,N’-methylene bisacrylamide (MBA) as the initiator and cross-linker, respectively. A cross-linked cellulose–poly(sodium acrylic acid) hydrogel (CE–PAANa) was used to remove Cu(II), Pb(II), and Cd(II) heavy metal ions from aqueous solutions and to explore the effects of adsorption conditions, such as the ratio of reactants, pH of the solution, and ionic strength, as well as to evaluate the absorption kinetics, equilibrium adsorption isotherms, and adsorption performance of the adsorbent. The surface structure and morphology of the hydrogels were characterised using scanning electron microscopy (SEM) and energy-dispersive X-ray spectrometry (EDS). Fourier-transform infrared spectroscopy (FTIR) and X-ray photoelectron spectroscopy (XPS) were used to analyse the adsorption mechanisms.

## 2. Results and Discussion

### 2.1. Characterisation

#### 2.1.1. Surface Morphology Analysis

The surface of the SCB fibres was smooth, and some non-cellulose cementing materials were present on the fibre surface (Figure 1a), which were probably fibre bundles formed by pectin, hemicellulose, lignin, and other impurities [31]. In Figure 1b, CE has a sheet-like appearance, exhibiting coarse micro-sized protofibrils owing to the structural decomposition of the fibrous chains into smaller-sized microcrystals during the acid hydrolysis of the glycosidic bonds of CE [32]. As shown in Figure 1c, the dehydrated CE–PAANa gel had a crumpled morphology. Figure 1d shows an open honeycomb 3D network structure formed by water absorption and swelling of the gel, which was due to the electrostatic rejection of a large number of carboxyl groups [33]. The CE-PAANa had a swelling capacity of 75 g/g in distilled water, as shown in Appendix A and Appendix A. This structure facilitates the diffusion of metal ions from the outside to the inside of the adsorbent, providing a larger specific surface area, more ligands for the adsorption of metal ions, and allowing the pollutants to be fully adsorbed [34].

The EDS spectra of CE–PAANa and CE–PAANa–Cu(II) displayed in Figure 2 and Figure 3, respectively, clearly show that the hydrogel contains C and O as the main skeleton. As shown in Figure 3e, the proportion of Cu was 30.07%, and a large amount of Cu(II) was adsorbed onto CE–PAANa. Cu was evenly distributed on the surface of CE–PAANa, indicating that Cu(II) was adsorbed onto the hydrogel through a layered approach, which is compatible with the Langmuir model of monolayer adsorption. The Na content decreases from 18.13% to 0.13% after the adsorption of Cu(II). This demonstrates that the Na ions in CE–PAANa were substituted by heavy metal ions through cationic exchange interaction [35], and that the adsorption process was chemisorption [13].

#### 2.1.2. X-ray Diffraction (XRD) Analysis

Figure 4 shows the XRD diffraction pattern of the polymeric material CE–PAANa. The crystallinity indices (*C.I.*) were derived from the crystalline and amorphous intensity values obtained from XRD [36] (Equation (2)):(1)$$\%C.I.=\frac{I_{200}-I_{am}}{I_{200}}\times 100$$

*I*_200_ is the maximum intensity peak at 2θ between 22° and 23° in the crystalline region, and *I_am_* is the intensity at 2θ between 15° and 18°. The *C.I.* of SCB, CE, and polymer was 48.75%, 70.87%, and 39.89%, respectively. The increase in *C.I*. during the SCB extraction-to-CE process was attributed to the removal of the amorphous region of hemicellulose, leading to the rearrangement of CE molecules. In contrast, the grafting and polymerisation of CE disrupted its crystallinity, leading to a decrease in *C.I.* As shown in Figure 4, the diffraction peaks at 12.5°, 21.9°, and 34.6° correspond to the 002, 101, and 040 crystalline planes of the CE, respectively [37]. The disappearance and weakening of the hydrogel diffraction peaks proved that graft polymerisation of CE–PAANa had occurred. The XRD pattern of the hydrogel showed a weak diffraction peak at 22.4°, indicating that grafting and polymerisation increased the amorphous nature of the polymer.

#### 2.1.3. FTIR Analysis

The FTIR spectra of various functional groups in SCB, CE, and CE–PAANa are shown in Figure 5a,b. For SCB and CE, the absorption band observed near 3414 cm^−1^ was associated with the O–H groups, and the sharp peak at 2904 cm^−1^ was attributed to the C–H stretching vibration [38]. The peak at 1737 cm^−1^ is due to C=O bond in the acetyl ester of hemicellulose, and that at 1246 cm^−1^ is attributed to the aryl bond in lignin. However, these peaks disappeared in the spectrum of CE, confirming the removal of hemicellulose and lignin and the successful extraction of CE. The FTIR spectrum of CE shows the typical characteristic peaks of cellulose, with a strong and broad band at 3500–3000 cm^−1^, which is the O–H bond from AA, indicating the successful graft polymerisation of CE and AA. The additional peaks at 1433 cm^−1^ and 1042 cm^−1^ correspond to the bending vibration and C–O–C stretching vibration of the –CH_2_ group in glucopyranose [39]. The strong peak at 1690 cm^−1^ is attributed to the stretching of the C=O group of polyacrylic acid [40].

Figure 5b shows that the peaks of the hydrogel after the adsorption of metal ions were almost the same except for the difference in displacement at the left and right. The C=O bond of CE–PAANa was located at 1690 cm^−1^. After adsorption of the heavy metal ions Cu^2+^, Pb^2+^, and Cd^2+^, the peaks shifted to 1711, 1706, and 1709 cm^−1^, respectively, because of C=O bond strengthening due to the formation of complexes with Cu^2+^, Pb^2+^, and Cd^2+^ ions [41,42]. In brief, the carboxyl group and heavy metal ions Pb^2+^, Cu^2+^, and Cd^2+^ undergo coordination, resulting in a shift of the absorption peaks to higher wave numbers after adsorption of the metal ions, which strongly supports the coordination of CE–PAANa and heavy metal ions. This suggests that the carboxyl groups are involved in the adsorption of pollutants.

#### 2.1.4. Thermogravimetric Analysis

The pyrolysis properties of the CE–PAANa hydrogels were analysed using TGA and DTG, as shown in Figure 6a,b, respectively. The first stage from room temperature to approximately 100 °C, was the water loss stage of the specimens, with a weight loss of 9.2%. However, the chemical composition of CE–PAANa barely changes during this stage. The second stage, approximately 100–420 °C, was the main pyrolysis stage. In this stage, the specimen mass loss was accelerated at a temperature of 282 °C leading to a mass reduction of 42.34%, and the first wave of the DTG curve. At this time, the CE decomposition at 295–346 °C was intense. The third stage (420–530 °C) was the carbonisation stage, which was mainly the slow decomposition of the CE pyrolysis residue and the decomposition of the carboxyl group of AA [43]. As a result, the mass was reduced by 27.38%. The residual mass of CE–PAANa is 20.1%. TG analysis revealed that the grafted AA chains formed a cross-linked structure on the CE [13], which enhanced its thermal stability.

#### 2.1.5. Adsorption Mechanism

The adsorption mechanism of CE–PAANa was explored using XPS. The survey spectrum clearly shows strong peaks for Cu 2p, Pb 4f, and Cd 3d (Figure 7a), which prove that Cu(II), Cd(II), and Pb(II) exist on the hydrogel surface. The characteristic peaks of Na 1s at 1071 eV disappeared after the adsorption of Cu(II), Cd(II), and Pb(II), which agrees well with the EDS results. This was mainly due to the ion exchange of the –COONa functional group with metal ions in solution [44], and the –COOH and –OH functional groups interacting with metal ions in coordination or electrostatically.

The presence of the Cu 2p_1/2_ and Cu 2p_3/2_ peaks in Figure 7b verifies that Cu^2+^ ions were adsorbed by CE–PAANa. Strong satellite peaks in the Cu 2p spectrum were observed at 944.36 eV and 963.1 eV, which could be attributed to copper oxide (CuO) [45]. The Pb 4f spectrum presented in Figure 7c is deconvoluted into four peaks; peaks at 143.3 eV and 144.1 eV correspond to Pb 4f_5/2_, whereas peaks at 138.6 eV and 139.2 eV correspond to Pb 4f_7/2_. The Pb 4f_7/2_ peaks could be attributed to PbCO_3_ and (Pb_3_(OH)_2_(CO_3_)_2_) [46,47], demonstrating the presence of Pb in the form of Pb-O and Pb^2+^. As shown in Figure 7d, two new peaks at 412.8 eV and 406.1 eV were attributed to Cd 3d_3/2_ and Cd 3d_5/2_, respectively, indicating that Cd existed as Cd^2+^ (Cd(OH)_2_) in the polymer [48].

As shown in Figure 7e, the O 1s spectrum could be deconvoluted into C–O, C = O, and O–H/COO^-^. It can be seen that the binding energies increased after the adsorption of Cu(II), Cd(II), and Pb(II), indicating the involvement of oxygen-containing functional groups in the adsorption process and the electrostatic interaction between the carboxylate group and targeted pollutants [44]. During the complexation process, the donation of lone pairs of electrons from O leads to a decrease in the electron density around the O atom, resulting in a larger binding energy [49]. Therefore, adsorption occurs mainly through the complexation of –COO^-^ with metal ions and the exchange of –COO^-^Na^+^ with the metal ions [50].

Based on the analysis of the adsorption mechanism, the adsorption of heavy metal ions was divided into the following steps: (1) Cu(II), Cd(II), and Pb(II) were rapidly complexed on the surface of the hydrogel for good adsorption; (2) through ion exchange, the original position of the sodium ions was replaced by heavy metal ions [51,52,53].

### 2.2. Adsorption Studies

#### 2.2.1. Adsorption Kinetics

Figure 8a shows the variation in the adsorption performance with contact time at an initial concentration of 200 ppm for Cu(II), Pb(II), and Cd(II). The results show that the hydrogel exhibited high adsorption rates during the initial adsorption phase and reached adsorption equilibrium after approximately 100 min. This is explained by the fact that the adsorbent can provide sufficient adsorption sites at the beginning of adsorption. The hydrogel gradually dissolves, resulting in the internal pores becoming larger, leading to faster diffusion of metal ions, and the overall performance gradually increasing. In the later stages, the main adsorption sites on the surface of the adsorbent were occupied, showing a saturated state, and the diffusion of metal ions slowed [54].

In the solid–liquid adsorption system, to study the effect of CE–PAANa on the adsorption rate, control mechanism of mass transfer, and chemical reactions of the adsorption process, the kinetic data were fitted to pseudo-first-order and pseudo-second-order kinetic models [55]:(2)$$\log(q_{e}-q_{t})=\log q_{e}-\frac{k_{1}}{2.303}t$$
(3)$$\frac{t}{q_{t}}=\frac{1}{k_{2}q_{e}^{2}}+\frac{t}{q_{e}}$$
where *q_e_* (mg/g) is the adsorption at equilibrium, *q_t_* (mg/g) is the adsorption at a specific time, and *k*_1_ (min^−1^) and *k*_2_ (g⋅mg/min) are the constants of the pseudo-first-order kinetic model and pseudo-second-order kinetic models, respectively.

As shown in Figure 8b,c and Table 1, the kinetic correlation coefficients of the pseudo-second-order kinetic models of the adsorbents for Cu(II), Pb(II), and Cd(II) (R^2^ = 0.9948, 0.9938, and 0.9939, respectively) are larger than those of the pseudo-first-order kinetic models (0.9794, 0.9767, and 0.9680, respectively). Therefore, the adsorption properties of CE–PAANa for Cu(II), Pb(II), and Cd(II) can be described by a pseudo-second-order kinetic model, indicating that the metal ions are controlled by chemisorption onto the adsorbent [56]. The adsorption rates of Cu(II), Pb(II), and Cd(II) onto CE–PAANa were governed by chemical processes [57]. These results suggest that the adsorption process may produce electron transfer, exchange, or shared formation of chemical bonds [58].

#### 2.2.2. Adsorption Isotherm

The effect of the initial metal ion concentration on the adsorbent performance was investigated (Figure 8d). As shown in the figure, the adsorption capacity, *q_e_*, increased gradually and saturated with an increase in the initial concentrations of Cu(II), Pb(II), and Cd(II). An increase in the initial concentration will accelerate the mass transfer rate of the liquid film on top of the adsorbent, which gradually tends to saturate.

The collected data were fitted to the linearised Langmuir and Freundlich isotherm models [59] to evaluate the adsorption behaviour of CE–PAANa. The Langmuir and Freundlich isotherm model relations are provided by Equations (5) and (6), respectively.
(4)$$\frac{C_{e}}{q_{e}}=\frac{C_{e}}{q_{m}}+\frac{1}{q_{m}k_{L}}$$
(5)$$\ln q_{e}=\ln k_{F}+\frac{1}{n}\ln C_{e}$$
where *C_e_* (mg/L) is the concentration of the metal ion solution at equilibrium, *q_e_* and *q_m_* are the equilibrium adsorption and maximum adsorption capacities, respectively, *k_L_* (L/mg) is the Langmuir constant, *k_F_* (L/mg) is the Freundlich constant, and *n* is the inhomogeneity factor.

The adsorption isotherms of CE–PAANa for Cu (II), Pb (II), and Cd (II) were obtained by nonlinear fitting, as shown in Figure 8e,f, the relevant parameters are summarised in Table 2. The results show that the Langmuir model is more suitable than the Freundlich model, with correlation coefficients of 0.9925, 0.9997, and 0.9912, respectively. The curve of *C_e_/q_e_* versus *C_e_* was perfectly linear (Figure 8e), and the theoretical *q_m_* values obtained from the Langmuir adsorption isotherm matched the experimental values well. The applicability of the Langmuir model proved that Cu (II), Pb (II), and Cd (II) were adsorbed as a monolayer onto the adsorbent [60]. The maximum adsorption amounts (q_m_) of Cu (II), Pb (II), and Cd (II) onto CE–PAANa were 106.3, 333.3, and 163.9 mg/g, respectively.

#### 2.2.3. Effect of pH on Adsorption

The pH value significantly impacts the adsorption performance of the adsorbent, which affects the surface charge of the adsorbent and the form of metal ions present in the wastewater [61]. Therefore, the effect of the solution pH (2.0–6.0) on the adsorption of Cu(II), Cd(II), and Pb(II) by the hydrogels was investigated. As shown in Figure 9a, the adsorption capacity of the adsorbent for all three metal ions increased significantly as the pH increased from 2 to 6, owing to the protonation of the adsorbent surface and the large number of hydrogen ions that compete for adsorption under acidic conditions, which is not conducive to the adsorption of metal ions [62]. Moreover, in a low-pH environment, the swelling capacity of the adsorbent is dramatically reduced, leading to a reduction in the heavy metal ions inside the spread hydrogel [63]. In addition, when the initial pH was greater than 6, the metal ions in the stock solution were present as hydroxides, leading to cloudy solutions and interference with parallel experiments. Therefore, all adsorption experiments were carried out at a pH of 5.

#### 2.2.4. Exploring Synthetic Ratios

Typically, the AA dose significantly affects the number of adsorption sites. As shown in Figure 9b, the best adsorption was observed for CE–PAANa with different AA, and the adsorption of Cu(II), Pb(II), and Cd(II) was the highest when the amount of AA was 8 g. The adsorption of Cu(II), Pb(II), and Cd(II) by pCE–PAANa synthesised with purchased CE was slightly higher than that of adsorbent (CE–PAANa) in this experiment. The CE extracted from SCB contained impurities, as shown in Table 3. The weight-average (Mw) molecular weight of CE was 9599 g/mol (Appendix A).

A comparison of the adsorption performance of CE–PAANa and other similar adsorbents is presented in Table 4. The results showed that CE–PAANa, which was synthesised by the simple radical polymerisation of grafted CE and AA, exhibited better adsorption ability.

#### 2.2.5. Stability and Reusability

Cyclic adsorption experiments are shown in Figure 9c. A 0.05 M EDTA solution was employed for CE–PAANa. The regeneration of CE–PAANa is a key indicator for evaluating adsorbent performance in terms of large-scale wastewater treatment. Recycling waste CE–PAANa can reduce costs and avoid secondary environmental pollution. The adsorption capacity decreased with each cycle. However, even after the third regeneration cycle, the hydrogel retained considerable Cu(II), Pb(II), and Cd(II) adsorption capacities. The adsorption capacities of CE–PAANa for Cu(II), Pb(II), and Cd(II) were 92.73%, 88.56%, and 82.09%, respectively.

## 3. Materials and Methods

### 3.1. Materials

SCB (CE raw material) was obtained from the Guangxi Nanning Farmers’ Market. Sodium hydroxide (NaOH 95%), hydrochloric acid (HCl 37%), hydrogen peroxide (H_2_O_2_ 30%), AA (98%). MBA (99%) and KPS (98%) were provided by the Shanghai Aladdin Biochemical Technology Co. Copper nitrate (Cu(NO_3_)_2_⋅3H_2_O), ethylenediaminetetraacetic acid disodium (EDTA), lead nitrate (Pb(NO_3_)_2_) and cadmium chloride (CdCl_2_) were supplied from the Xilong Chemical Co. All chemicals were of reagent grade, and all aqueous solutions were prepared using distilled water.

### 3.2. Extraction of CE

The extraction of CE from an SCB was achieved mainly by the alkaline hydrogen peroxide technique [32], as shown in Figure 10. First of all, the SCB was repeatedly washed, dried, placed into a pulveriser at 34,000 r/min for 30 s, and then sieved through a 400 μm mesh sieve. It was boiled with alkali in 10% (*w/v*) NaOH solution (80 °C for 4 h) to remove organic macromolecular impurities such as lignin and hemicellulose. The filter residue was then transferred to a bottle with 2% (*w/v*) NaOH solution and bleached with 15% (*v/v*) H_2_O_2_ at 60 °C for 2 h. The mixture was then filtered with a Buchholz funnel and washed repeatedly with distilled water. The extracted solids were reacted with 4 M HCl at 80 °C for 1 h. After filtering and washing until the pH was neutral, the solution was dried to obtain CE.

### 3.3. Preparation of CE–PAANa

One gram of CE and 40 mL of distilled water were combined, prepared as a suspension, and transferred to a 250 mL three-neck flask after 10 min of sonication. Then, 0.3 g of a KPS initiator was added. A mixture of 8 g of AA (neutralised with 9 g of 13% (*w/w*) NaOH) and 30 mg of MBA was added to a constant-pressure dropping funnel. The amounts of different AA (4, 6, 8, and 10 g) were explored. The three-neck flask was assembled, and the obtained solution was purged with N_2_ to remove the dissolved oxygen. The solution was stirred at 50 °C in a constant-temperature water bath for 20 min. Subsequently, the monomer AA and cross-linker were added dropwise for 30 min. Finally, the mixture was stirred at 60 °C for 1 h to complete the polymerisation. The samples were cut into small slices of a specific size and repeatedly washed with 20% ethanol and distilled water. Finally, CE–PAANa were placed on a surface dish and dried in a drying oven at 60 °C for constant weight. Schematic diagram of the structure of CE-PAANa, as shown in Appendix A.

### 3.4. Characterisation

The functional groups of the CE copolymers were identified using FTIR (NICOLET-6700, Thermo, New York, USA) in the range of 4000–500 cm^−1^. Thermogravimetry (SDT Q600) was carried out with a heating rate of 15 °C/min under N_2_ flow in the range of 25–600 °C). The morphology and composition of the samples were observed by SEM (HITACHI SU5000, Hitachi High-Tech, Tokyo, Japan) and EDS (INCA MAX-50, Oxford Instrument Technology, Oxford, UK). XPS (ESCAL-AB-250XI, Thermo, New York, NY, USA) was used to further characterise the elemental composition of the samples and analyse the chemical state of the sample surface before and after adsorption. The crystal structures of the samples were determined using X-ray powder diffraction (XRD, Mini Flex 600, Rigaku, Tokyo, Japan). CE, lignin, and hemicellulose were measured using a cellulose analyser (ANKOM 220, Ankom, Qingdao, China) and the paradigm washing method.

### 3.5. Adsorption Experiments

Cu(II), Cd(II), and Pb(II) were selected as the heavy metal ions in this study. Adsorption experiments were performed using 50 mg of CE–PAANa in 50 mL of 200 mg/L Cu(II), Cd(II), and Pb(II) solutions at room temperature (24 °C). The pH varied from 1 to 6, the reaction time was 6 h, and the initial concentration was 200 mg/L. The pH values of the Pb(II) solutions were adjusted with a 0.1 M HNO_3_ or NH_3_–H_2_O solution. The adsorption kinetics was set from 0 to 300 min (pH 5 and C_0_ = 200 mg/L). The adsorption performance of CE–PAANa for heavy metal ions was examined at different initial concentrations (50–700 mg/L), pH 5.0, and an equilibration time of 2 h.

Parallel experiments were performed in a 180 rmp thermostatic shaker (HY-5B, Xinrui Instrument Co., Ltd., Changzhou, China). After adsorption, the final concentration of heavy metal ions in the bottle was analysed using inductively coupled plasma mass spectrometry (ICP-MS, Opetima 7000 DV, PerkinElmer, Waltham, MA, USA). The adsorption capacity, *q_e_* (mg/g), was quantified using Equation (1):(6)$$q_{e}=\frac{C_{0}-C_{e}}{m}v$$
where *C*_0_ and *C_e_* represent the initial and equilibrium concentrations (mg/L), respectively. V is the volume of the initial solution (L), and m is the weight of CE–PAANa (g).

## 4. Conclusions

In this study, CE extracted from SCB was graft-copolymerised with an AA monomer to CE–PAANa using simple free radical polymeriaation and applied to the adsorption of heavy metal ions. The hydrogel has a porous honeycomb structure, which can provide a large number of adsorption sites for metal ions. XPS and EDS showed that the adsorption mechanisms of Cu(II), Pb(II), and Cd(II) were mainly through the complexation of –COO^-^ with metal ions and the exchange of –COO^-^Na^+^ with metal ions. Therefore, the carboxyl group on CE–PAANa significantly affects the adsorption of metal ions. The pH, contact time, and initial concentration of Cu(II), Pb(II), and Cd(II) affected the adsorption performance of the adsorbents. The adsorption process is consistent with the pseudo-second-order kinetic model and the Langmuir isothermal model, and there is a chemical interaction between the metal ions and the hydrogel. The maximum adsorption capacities of CE–PAANa for Cu(II), Pb(II), and Cd(II) were 106.3, 333.3, and 163.9 mg/g, respectively. Although CE-PAANa has high adsorption capacities for heavy metal ions, future work will be required to explore a multitude of mixed metal ions in order to simulate the actual wastewater environment. In summary, the CE–PAANa hydrogel was shown to have a low cost, high heavy metal adsorption capacity, and reusability. The potential development of biomass waste SCB for heavy metal wastewater treatment contributes to environmental and economic sustainability.

## Figures and Tables

**Figure 1 ijms-24-08922-f001:**
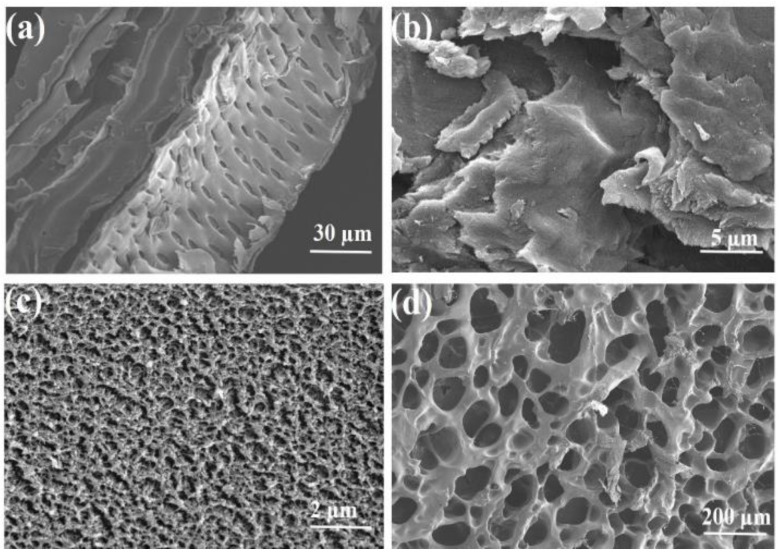
SEM images of (**a**) SCB, (**b**) CE, (**c**) dried hydrogel of CE–PAANa, and (**d**) CE–PAANa after swelling.

**Figure 2 ijms-24-08922-f002:**
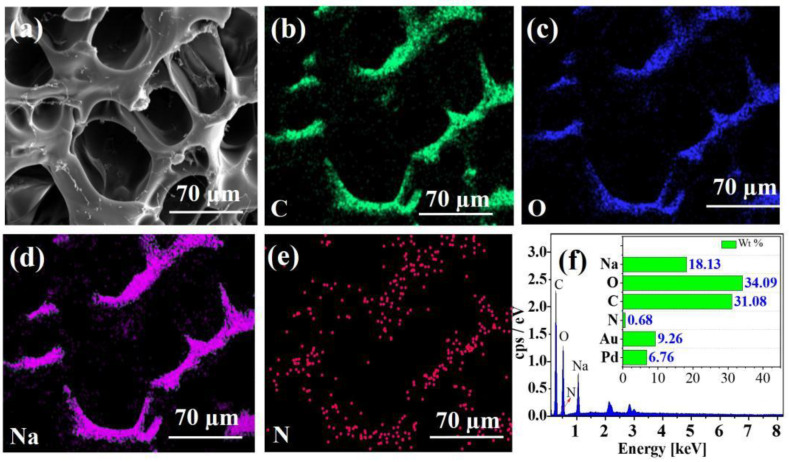
(**a**) SEM image of CE–PAANa, (**b**–**e**) corresponding EDS map with selected elements (C, O, Na, and N) before adsorption, (**f**) EDS curves of CE–PAANa.

**Figure 3 ijms-24-08922-f003:**
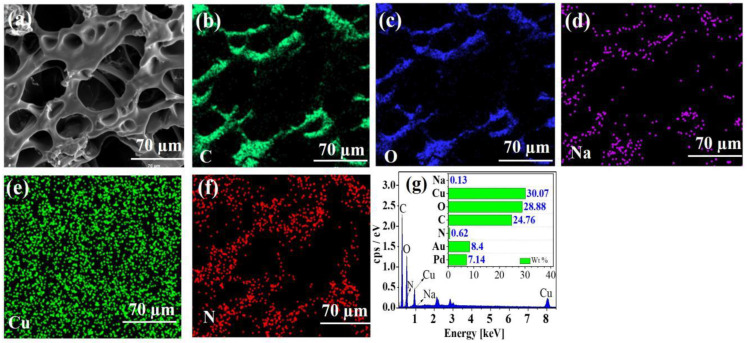
(**a**) SEM image of CE–PAANa–Cu(II), (**b**–**f**) corresponding EDS map with selected elements (C, O, Na, Cu, and N) after adsorption. (**g**) EDS curves of CE–PAANa–Cu(II).

**Figure 4 ijms-24-08922-f004:**
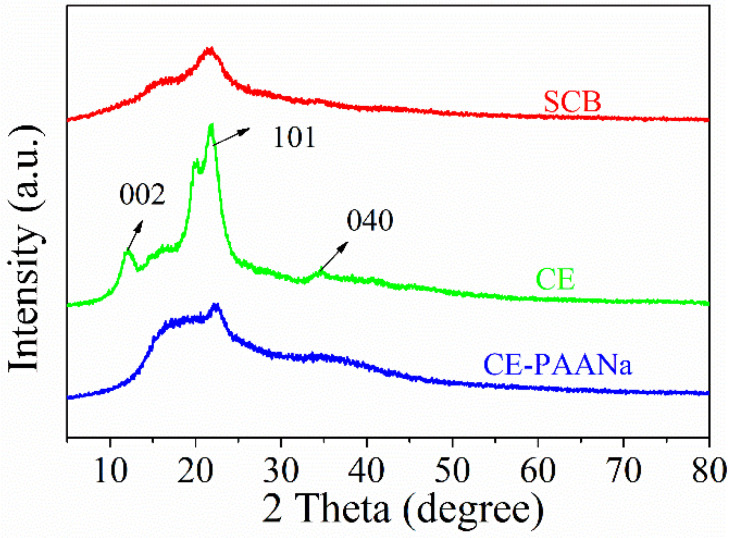
XRD curves of SCB, CE, and CE–PAANa.

**Figure 5 ijms-24-08922-f005:**
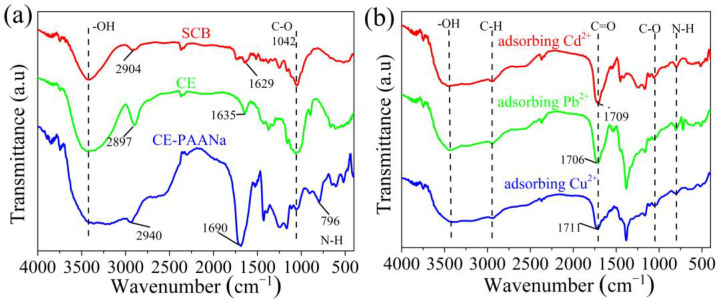
(**a**) FT–IR spectra of SCB, CE and CE–PAANa, (**b**) FTIR spectra of CE–PAANa after adsorption of Pb(II), Cd(II), and Cu(II).

**Figure 6 ijms-24-08922-f006:**
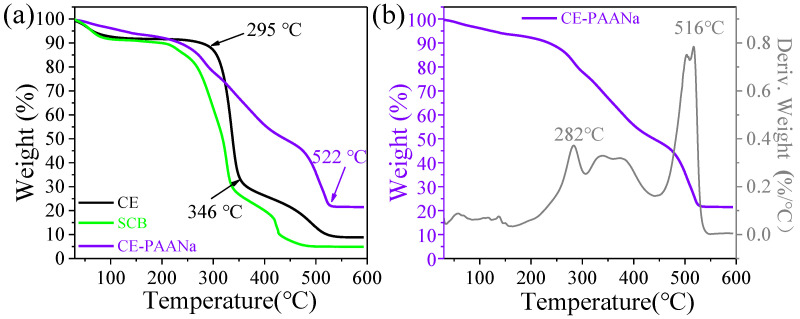
(**a**) TGA curves of CE, SCB, and CE–PAANa; (**b**) DTG curves of CE–PAANa.

**Figure 7 ijms-24-08922-f007:**
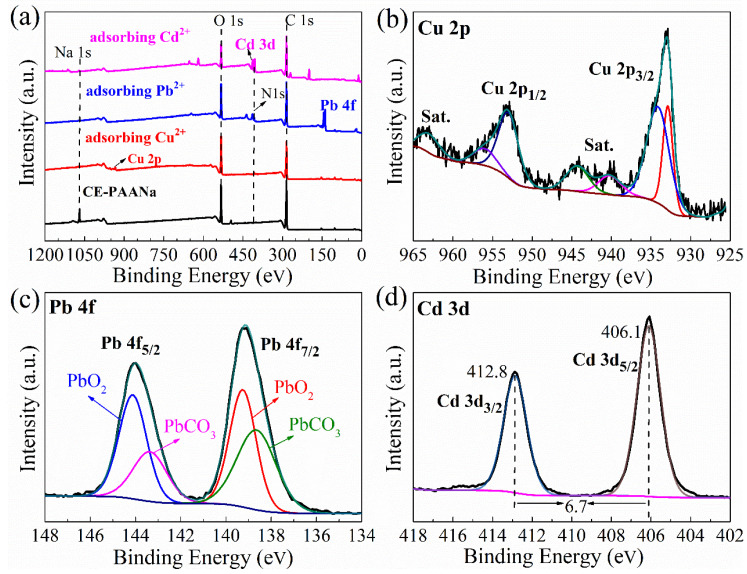
(**a**) XPS survey of the CE–PAANa before and after adsorption; (**b**–**d**) XPS spectra region of Cu 2p, Pb 4f, and Cd 3d. (**e**) O 1 s spectra of CE–PAANa before and after the adsorption of Cu(II), Cd(II), and Pb(II).

**Figure 8 ijms-24-08922-f008:**
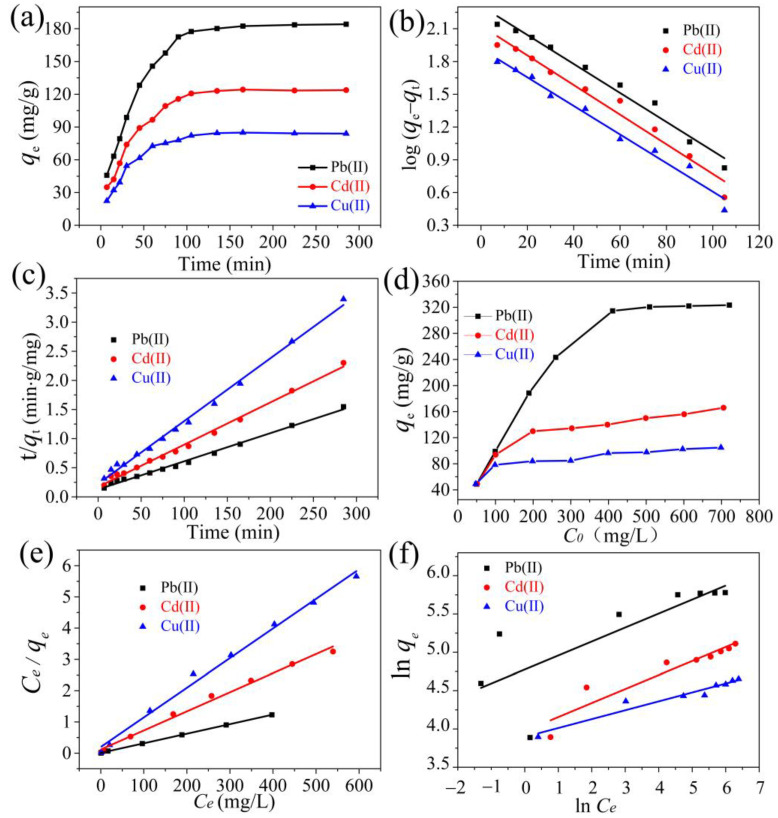
(**a**) Contact time of Cu(II), Cd(II), and Pb(II) adsorption by CE–PAANa; (**b**) pseudo-first-order and (**c**) pseudo-second-order kinetics; (**d**) initial metal ion concentrations of Cu(II), Cd(II), and Pb(II) adsorption by CE–PAANa; (**e**) Langmuir and (**f**) Freundlich isotherms, respectively.

**Figure 9 ijms-24-08922-f009:**
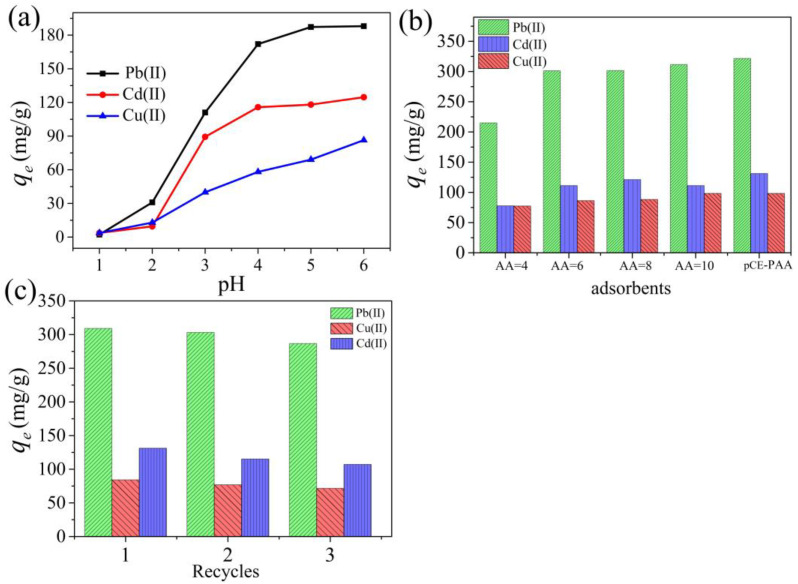
(**a**) Effect of pH of CE–PAANa; (**b**) CE-PAANa at various doses of AA and pCE–PAANa; (**c**) adsorption of Pb(II), Cd (II), and Cu (II) by CE–PAANa during three regeneration cycles.

**Figure 10 ijms-24-08922-f010:**
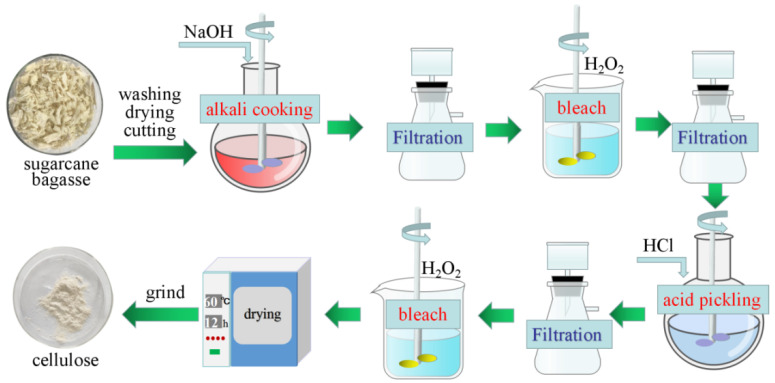
Schematic illustration of CE preparation from SCB.

**Table 1 ijms-24-08922-t001:** Fitting parameters of pseudo-first-order and pseudo-second-order kinetic models for Pb(Ⅱ), Cd(Ⅱ), and Cu(Ⅱ) adsorption onto CE-PAANa.

Adsorbent	Pseudo-First-Order Model			Pseudo-Second-Order Model		
	K_1_ (min^−1^)	q_e_ (mg/g)	R^2^	K_2_ (min^−1^)	q_e_ (g⋅mg/min)	R^2^
Pb (Ⅱ)	0.03058	203.4	0.9767	1.813 × 10^−4^	207.0	0.9938
Cd (Ⅱ)	0.03125	134.1	0.9680	3.061 × 10^−4^	137.9	0.9939
Cu (Ⅱ)	0.03015	82.55	0.9794	5.178 × 10^−4^	92.76	0.9948

**Table 2 ijms-24-08922-t002:** Fitting parameters of Langmuir and Freundlich models for Pb(II), Cd(Ⅱ), and Cu(Ⅱ) adsorption onto CE–PAANa.

Adsorbent	Langmuir Model			Freundlich Model		
	K_L_ (L/g)	q_max_ (mg/g)	R^2^	K_F_ (L/g)	1/n	R^2^
Pb (Ⅱ)	0.2971	333.3	0.9997	118.6	0.1825	0.5632
Cd (Ⅱ)	0.05209	163.9	0.9912	53.14	0.1827	0.8743
Cu (Ⅱ)	0.04731	106.3	0.9925	49.24	0.1160	0.9373

**Table 3 ijms-24-08922-t003:** Main components of SCB and extracted CE.

Types	Cellulose %	Hemicelluloses %	Lignin %	Yield %
SCB	33.75	30.77	13.16	-
CE	77.83	6.11	2.18	30.12

**Table 4 ijms-24-08922-t004:** Comparison of ions removal by CEPAANa by other similar composites.

Adsorbent	q_m_ (mg/g)			pH	T(k)	Refs
	Cu^2+^	Pb^2+^	Cd^2+^			
PVA/PAA gel	-	195.0	115.9	5	313	[64]
Polyampholyte hydrogel	-	216.1	153.8	Pb 5.0 Cd 6.0	313	[65]
Cellulose nanocrystal-g-poly(acrylic acid-co-acrylamide) aerogels	-	366.3	-	6	293	[66]
GAMAAX	-	-	312.1	7	298	[67]
P(AANa-co-AM)/GO hydrogel	-	452.3	196.4	4.5	298	[50]
MCC-g-(AA-co-AM)	157.5	393.2	289.9	5	300.1	[13]
Cellulose-g-poly-(acrylamide-co -acrylic acid) polymeric	61.8	209.6	101.7	Cu Pb 5.5 Cd 6.0	298	[17]
CE–PAANa	106.3	333.3	163.9	5	296.6	this work

## Data Availability

Data sharing is not applicable.

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
