# Peer review of "Synthesis of Cellulose–Poly(Acrylic Acid) Using Sugarcane Bagasse Extracted Cellulose Fibres for the Removal of Heavy Metal Ions"

_ijms, 2023, doi:10.3390/ijms24108922_

Round 1

Reviewer 1 Report

After a careful rereading of the manuscript, I believe it needs strong editing to be accepted in a scientific journal. The topic covered is interesting and extremely actual, so I would suggest that the authors improve the content of the paper from a stylistic point of view (a revision of English is necessary) and the exposition of the scientific content. The introduction should be better developed by adding more literature references that would better contextualize the topic. The Materials and Methods part needs to be more detailed, particularly in the material's synthesis. In the Results and Discussion part, reference is made to missing images (Figure 2 is not present), and discussions of the results are often unclear. Given the large amount of data collected and analysis performed, I would suggest a complete revision of the paper and a future resubmission of the paper, considering the interest in the topic covered.

Author Response

After a careful rereading of the manuscript, I believe it needs strong editing to be accepted in a scientific journal. The topic covered is interesting and extremely actual, so I would suggest that the authors improve the content of the paper from a stylistic point of view (a revision of English is necessary) and the exposition of the scientific content.

Reply: In order to accurately introduce the research content of this manuscript, we carefully read the full text and corrected some language and writing problems, and marked the changes in red. We really hope that the language level have been substantially improved.

The introduction should be better developed by adding more literature references that would better contextualize the topic.

Reply: We thank the reviewers for the constructive and very helpful comments. In our resubmitted manuscript, we have added more literature references and marked the changes in red.

The Materials and Methods part needs to be more detailed, particularly in the material's synthesis.

Reply: We thank the reviewers for the constructive and very helpful comments. In the revised manuscript, we have modified the material's synthesis (Lines 107, 109, and 110 on page 3).

In the Results and Discussion part, reference is made to missing images (Figure 2 is not present), and discussions of the results are often unclear.

Reply: We were really sorry for our mistakes. We corrected this error, we have added Figure 2 in the revised manuscript (Lines 148 on page 4).

Reviewer 2 Report

The manuscript entitled "Synthesis of Cellulose-Poly(acrylic acid) Using Sugarcane Bagasse Extracted Cellulose Fibers for the Removal of Heavy Metal Ions" presents a novel approach for the removal of heavy metal ions using cellulose-poly(acrylic acid) synthesized from sugarcane bagasse extracted cellulose fibers. The manuscript is well written, and the methodology is well described, making it easy to understand the steps taken in the synthesis of the cellulose-poly(acrylic acid).

The results presented in the manuscript indicate that the synthesized cellulose-poly(acrylic acid) was effective in the removal of heavy metal ions from aqueous solutions. The authors have also discussed the mechanism involved in the removal of heavy metal ions, which is an  advantage.

Figure 2 is not present in the manuscript, and only the corresponding caption is provided.

Overall, this manuscript presents an interesting approach for the removal of heavy metal ions using cellulose-poly(acrylic acid) synthesized from sugarcane bagasse extracted cellulose fibers.This manuscript has the potential to make a valuable contribution to the field of environmental remediation.

Author Response

The manuscript is well written, and the methodology is well described, making it easy to understand the steps taken in the synthesis of the cellulose-poly (acrylic acid).

The results presented in the manuscript indicate that the synthesized cellulose-poly (acrylic acid) was effective in the removal of heavy metal ions from aqueous solutions. The authors have also discussed the mechanism involved in the removal of heavy metal ions, which is an advantage.

Figure 2 is not present in the manuscript, and only the corresponding caption is provided. Overall, this manuscript presents an interesting approach for the removal of heavy metal ions using cellulose-poly (acrylic acid) synthesized from sugarcane bagasse extracted cellulose fibers. This manuscript has the potential to make a valuable contribution to the field of environmental remediation.

Reply: We thank the Reviewer 2 for the positive assessment of our manuscript. In the revised manuscript, we have added Figure 2 (Lines 148 on page 4).

Reviewer 3 Report

I studied the manuscript concerning "Synthesis of cellulose-poly(acrylic acid) using sugarcane bagasse and extracted cellulose fibers for the removal of heavy metal ions" with great interest. The cellulose was successfully extracted from the SCB, and the synthesized cellulose-based hydrogel was synthesized via simplistic free radical graft polymerization. Furthermore, various influencing factors on the batch adsorption capacity were investigated. Finally, authors confirmed that the results demonstrated that the developed cellulose graft copolymer sorbents could be potentially used for the removal of heavy metal ions such as Cu(II), Pb(II), and Cd(II).

In my opinion, the idea of this manuscript is good. However, this paper contains some limitations that are not discussed or even mentioned and should be clarified. My concerns and comments are outlined below.

1.       Why did authors use only one concentration of cellulose (one gram of CE and 40 ml of distilled water, etc.)? Other cellulose concentrations should be used in this study.

2.      What exactly is the monomer AA? Please provide some information concerning this monomer as well as the abbreviation AA.

3.      The authors should determine the Mw of the extracted cellulose from the SCB.

4.      The authors should carry out some swelling experiments to confirm that the developed cellulose graft copolymer sorbent is considered a hydrogel.

5.      There are some issues with respect to sample sizes and statistical considerations that need to be addressed. How many measurements were made per condition? What is the significance of the difference between the results?

6.      Tables 1, 2, and 3: The values of these should be presented as mean +/- standard deviation.

7.      Figure 10: The graphical representations should be presented as mean +/- standard deviation and include the significance level difference (*p) between samples and controls.

8.      The authors should update the Figure 4 caption. Please provide the right Figure 4G caption as well as the right Figure 4F caption since it is not EDS curves of CE-PAANa-Cu(II).

9.      The authors should discuss whether or not there is an interpenetration network between cellulose and the monomer AA as a function of the concentration used.

10.   The discussion should be enriched by explaining the chemistry of the developed hydrogel.

11.   This paper needs comments on the limitations and perspective at the end of this study.

Author Response

In my opinion, the idea of this manuscript is good.

Reply: We thank the Reviewer 3 for the positive assessment of our manuscript.

  1. Why did authors use only one concentration of cellulose (one gram of CE and 40 ml of distilled water, etc.)? Other cellulose concentrations should be used in this study.

Reply: We thank the Reviewer for raising of concerns, indeed, this is an important question. Cellulose and water were formed as a suspension, and the final graft with cellulose was AA. We reviewed the literature (Chen, Y.; Li, Q.; Li, Y.; Zhang, Q.; Huang, J.; Wu, Q.; Wang, S., Fabrication of Cellulose Nanocrystal-g-Poly(Acrylic Acid-Co-Acrylamide) Aerogels for Efficient Pb(II) Removal. In Polymers, 2020; Vol. 12.), the amount of cellulose was not significantly influenced the CE-PAANa adsorption performance. Therefore, we refer to the literature (Guleria, A.; Kumari, G.; Lima, E. C., Cellulose-g-poly-(acrylamide-co-acrylic acid) polymeric bioadsorbent for the removal of toxic inorganic pollutants from wastewaters. Carbohydrate Polymers 2020, 228, 115396), this experiment explored the effect of the amount of AA on the adsorption performance in the presence of sufficient amount of cellulose (Fig 10b).

  1. What exactly is the monomer AA? Please provide some information concerning this monomer as well as the abbreviation AA.

Reply: We would like to thank the reviewers for pointing out this important detail. Monomer AA is an abbreviation for acrylic acid. We have described this in revised manuscript (Lines 71 on page 2).

  1. The authors should determine the Mw of the extracted cellulose from the SCB.

Reply: We thank the reviewers for spotting this, Table S1 in the supplementary material showed that the Mw of the extracted cellulose from the SCB was 9599 g/mol.

Table S1. Gel Permeation Chromatography (GPC) test results for cellulose.

Sample

Method

Retention time 

(min)

GPC M

(g/mol)

GPC Mw (g/mol)

GPC Mz (g/mol)

GPC Mz+1 (g/mol)

GPC Mη (g/mol)

Dispersity

 Mw/Mn

CE

DMSO-

PMMA

8.6471

6678

9599

12973

16415

9134

1.4374

  1. The authors should carry out some swelling experiments to confirm that the developed cellulose graft copolymer sorbent is considered a hydrogel.

Reply: We thank the reviewers for the constructive and very helpful comments. Swelling experimental data has been in the supplementary material, the CE-PAANa had swelling capacity of 75 g/g in distilled water.

Fig. S2. Swelling ratios of CE-PAANa

  1. There are some issues with respect to sample sizes and statistical considerations that need to be addressed. How many measurements were made per condition? What is the significance of the difference between the results?

Reply: We thank the reviewer for this very insightful comment. ICP-OES automatically detected heavy metal ions three times and generated the average value, in this experiment the raw data were average values; Differences in measurement results can indicate issues such as instrument stability, sample homogeneity and data reliability.

  1. Tables 1, 2, and 3: The values of these should be presented as mean +/- standard deviation.

Reply: We kindly thank the Reviewer for the suggestion. The values generated by fitting the mean value according to Eq 3-6 in Tables 1, 2. SCB and CE main components were measured only once in Table 3. So we couldn't present it.

  1. Figure 10: The graphical representations should be presented as mean +/- standard deviation and include the significance level difference (*p) between samples and controls.

Reply: Thanks for your suggestion. Since the data recorded in this experiment are averages, the results of the three times of ICP tests were not recorded. So it is unable to add error bars to the graph.

  1. The authors should update the Figure 4 caption. Please provide the right Figure 4G caption as well as the right Figure 4F caption since it is not EDS curves of CE-PAANa-Cu(II).

Reply: We were really sorry for our careless mistakes. We corrected this error in the revised manuscript (Lines 167 on page 5).

  1. The authors should discuss whether or not there is an interpenetration network between cellulose and the monomer AA as a function of the concentration used.

Reply: This maybe a problem worthy of research, we thank the Reviewer for this interesting question. Interpenetrating polymer networks (IPN) are dense polymers made of interlaced and interpenetrating macromolecular networks. The CE suspension prepared in this experiment utilized only sonic dispersion and did not dissolve the cellulose (the molecular chains of cellulose were not opened).

Cellulose are only grafted together by surface hydroxyl groups with AA, and the internal molecular chains are not cross-linked with AA (Fig. S1).

We have reviewed the relevant literature, the synthesis of cellulose-based IPNs must use dimethyl sulfoxide (DMSO) to dissolve cellulose (Maleki, L.; Edlund, U.; Albertsson, A.-C., Synthesis of full interpenetrating hemicellulose hydrogel networks. Carbohydrate Polymers 2017, 170, 254-263), or form homogeneous cellulose nanofiber dispersions by high pressure , etc (Ma, H.; Zhao, J.; Liu, Y.; Liu, L.; Yu, J.; Fan, Y., Controlled delivery of aspirin from nanocellulose-sodium alginate interpenetrating network hydrogels. Industrial Crops and Products 2023, 192, 116081) (Zhang, H.; Sun, X.; Hubbe, M. A.; Pal, L., Highly conductive carbon nanotubes and flexible cellulose nanofibers composite membranes with semi-interpenetrating networks structure. Carbohydrate Polymers 2019, 222, 115013).

Fig. S1. Schematic diagram of the structure of CE-PAANa

  1. The discussion should be enriched by explaining the chemistry of the developed hydrogel.

Reply: We kindly thank the Reviewer for the suggestion. We have analysed the mechanism of heavy metal ion adsorption by EDS and XPS. It suggestsed that the carboxyl groups are were involved in the adsorption processes of pollutants by FTIR. Figure 10c showed the cyclic adsorption performance of this hydrogel for testing. In addition, we have revised our conclusions.

 “The hydrogel has a porous honeycomb structure, which can provide a large number of adsorption sites for metal ions, and it has a swelling rate of 75 g/g. XPS and EDS showed that the adsorption mechanism of Cu(II), Pb(II) and Cd(II) is mainly through the complexation of -COO- with metal ions and the exchange of -COO-Na+ with metal ions.”  

  1.  This paper needs comments on the limitations and perspective at the end of this study.

Reply: We kindly thank the Reviewer for the suggestion. The adsorption was only performed for single metal ions, not for heavy metal ions of complex water quality.

In the revised manuscript, we have added limitations and perspective to the conclusion:

 “Although CE-PAANa has a high adsorption capacities for heavy metal ion, future work will be required to explore a multitude of mixed metal ions in order to simulate the actual wastewater environment. In summary, the CE–PAANa hydrogel was shown to have a low cost, high heavy metal adsorption capacity, and reusability”

Round 2

Reviewer 1 Report

After a re-reading of the draft, I can notice that the authors have done extensive editing work on the manuscript and it is now much improved. 

I recommend one last spell check (I happened to cross out incomplete sentences (line 183) but overall I think this work can be accepted due to the considerable improvement of the manuscript. 

Author Response

After a re-reading of the draft, I can notice that the authors have done extensive editing work on the manuscript and it is now much improved. 

I recommend one last spell check (I happened to cross out incomplete sentences (line 183) but overall I think this work can be accepted due to the considerable improvement of the manuscript. 

Reply: We would like to thank the reviewer 1 for pointing out this important detail. In the revised manuscript, we have modified the sentence (Line 183) for “The crystallinity indices (C.I.) were derived from the crystalline and amorphous intensity values obtained from XRD”.

Reviewer 3 Report

The authors have improved the manuscript quality as requested.

Author Response

The authors have improved the manuscript quality as requested.

Reply: We thank the Reviewer 3 for the positive assessment of our manuscript.